# A survey-wide association study to identify youth-specific correlates of major depressive episodes

**Rahul Madhav Dhodapkar**[ID]*

School of Medicine, Yale University, New Haven, Connecticut, United States of America

* rahul.dhodapkar@yale.edu

## Abstract

### Background

Major depressive disorder is a common disease with high mortality and morbidity worldwide. Though peak onset is during late adolescence, the prevalence of major depressive disorder remains high throughout adulthood. Leveraging an association study design, this study screened a large number of variables in the 2017 National Survey on Drug Use and Health to characterize differences between adult and youth depression across a wide array of phenotypic measurements.

### Methods

All dichotomous variables were manually identified from the survey for association screening. Association between each dichotomous variable and past-year major depressive episode (MDE) occurrence was calculated as an odds ratio for adults (≥18 years) and youth (12–17 years), and tested for significance with Fischer's exact test. Logarithm of the calculated odds ratios were plotted and fitted to a linear model to assess correlation between adult and youth risk factors.

### Results

Many of the screened variables showed similar association between past-year depressive episode occurrence in youth and adults; Lin's concordance correlation coefficient between adult and youth associations was 0.91 (95% CI 0.89–0.92). Differentially associated variables were identified, tracking: female sex, alcohol use, cigarette use, marijuana use, Medicaid/CHIP coverage, cognitive changes due to a mental, physical or emotional condition, and respondents' identification of a single depressive event as the worst experienced.

### Conclusions

While some youth-specific correlates of major depressive episodes were identified through screening, including some novel associations, most examined variables showed similar association with youth and adult depression. Further study of results is warranted, especially

**Data Availability Statement:** Part of the data underlying this study are from the 2017 National Survey on Drug Use and Health. The data can be found here: https://www.datafiles.samhsa.gov/

study-dataset/national-survey-drug-use-and-health-2017-nsduh-2017-ds0001-nid17939. The authors did not have special access privileges. The other data underlying the results of this study are available from Github using the following link: https://github.com/rahuldhodapkar/YouthDepressionSWAS.

**Funding:** This work is supported in part by the Riva Ariella Ritvo endowment at the Yale Child Study Center, and in part by the Irving Harris Foundation.

**Competing interests:** The authors have declared that no competing interests exist.

concerning the finding of increased association between marijuana use and depressive episodes in youth.

## Introduction

Depressive disorders are ubiquitous and constitute a substantial source of morbidity and mortality in the United States and other developed nations; almost every health care provider will encounter a patient with some form of depression during his or her career [1–3]. By a recent World Health Organization (WHO) estimate, depressive disorders affect more than 264 million people worldwide and are among one of the leading causes of non-fatal health loss as measured by years lost to disability [4]. Many of the affective disorders have an average age of onset during adolescence, but continue to comprise a large portion of psychiatric pathology throughout adulthood [5, 6]. Developing a better understanding of this class of diseases and how they evolve over the lifespan remains an important goal with implications for prognosis and treatment of patients.

While the Diagnostic And Statistical Manual Of Mental Disorders, Fifth Edition (DSM-5) does not separate major depressive disorder in adolescents from that in adults, some differences have been described between depressive disorders in these age cohorts. In the DSM-5 itself, it is noted that children and adolescents may present clinically with primarily irritable, rather than depressed, mood as is often seen in adults [7]. Other studies have identified further differences between adolescent and adult depression, including more behavior disturbances and less hypersomnia in adolescents [8]. Psychopharmacologic effectiveness also differs between adult and youth depression, with one systematic review of selective serotonin reuptake inhibitors (SSRIs) in the treatment of major depression in children in adolescents finding that fluoxetine alone showed consistent efficacy, while most agents have demonstrated effect in adults [9]. Finally, some recent genetic results report that genome-wide significant risk variants differ between adult- and earlier-onset major depressive disorder, with earlier-onset depression having a genetic architecture more similar to that of schizophrenia and bipolar disorder [10].

Evidence for differences in the presentation, treatment, and genetics of major depressive disorder in adults and youth justifies further examination of the nosology of the disorder in the context of patient age. Psychopathological classification has evolved significantly over the past century, and refinement continues as new observations are integrated into existing knowledge [11]. Major depression, like most psychiatric disease, is deeply embedded in the psychological, social, and developmental context in which it occurs. Targeted efforts examining age-dependent biopsychosocial factors in depression have yielded durable information about risk factors for depression. For example, adverse childhood experiences (ACEs) and social support have both been reported to modify risk of developing a depressive disorder [12, 13]. However, the inherent complexity of social and psychological factors remains a major challenge facing these targeted studies examining depressive disorders. Traditional hypothesis-testing approaches require a fairly narrow a priori definition of scope, often leveraging investigator experience and qualitative research to identify themes worth fleshing out. To date, quantitative approaches to screening for factors of potential interest remain underdeveloped in the field of depressive disorders.

This study uses an association study design to identify targets of interest for further investigation of the differentiation between youth and adult major depression. Association studies

have been widely used in other fields, especially in genetics, and employ a simple study design. At a high level, association studies examine the relative frequencies of a given variable in cases versus controls, and express these frequencies as an odds ratio with some confidence interval. In genetics, genome-wide association studies have provided compelling insight by scanning large amounts of data for effects, and best practices for such studies have been well developed [14]. This study uses past-year self-reported major depressive episode (MDE) as the measure of case-ness, and calculates associations "survey-wide" for each dichotomous variable in the 2017 National Survey on Drug Use and Health (NSDUH). This study screened only dichotomous variables to streamline statistical analysis within the association study framework, but in the future this approach might be expanded to include additional data types such as continuous time or Likert-type variables. While hypothesis-driven case-control studies carefully control for confounding and limit the number of exposures examined, the author's association study design hopes to provide a broader, albeit less granular, picture of disease-associated phenotypes to inform hypothesis testing. This method serves as a low-cost, logistically expedient adjunct to hypothesis-driven studies; the author's analytical code and variable type classification have been made freely available and are ready for adaptation by other researchers.

The overall goal of this study was to increase the breadth of age-dependent characterization of major depression using large-scale association analysis. The hope is that using this design would better leverage the significant data volume available in public survey datasets for the analysis of phenotypic diversity across a wide range of variables. To the author's knowledge, this study represents the first large-scale association study of the youth-specific correlates of major depressive episodes. This analysis was able to reproduce some youth-specific associations that have been previously described in the literature, and identify some new ones. Additionally, this study was able to provide a quantitative benchmark for the degree of concordance between measured drug use and mental health parameters in a survey between adult and youth with past year MDE.

## Methods

### Description of study population and data

The National Survey on Drug Use and Health (NSDUH) is directed annually by the Substance Abuse and Mental Health Services Administration (under the U.S. Department of Health and Human Services). The survey administrators randomly select household addresses and professional interviewers are used to collect data to ensure that results are high quality and representative of the United States population [15]. This study used the 2017 version of the survey [16], which collected 2,668 variables from 56,275 participants across 50 states and the District of Columbia (Table 1). The survey included both youths (aged 12–17 years) and adults (aged 18 + years) in its direct data collection. To access the survey data programmatically, the author created a specification file identifying all dichotomous variables in the survey as an input for downstream analysis. Occurrence of past-year MDE was identified on the NSDUH if a respondent reported having at least 5 out of the 9 symptomatic criteria (Table 1), where at least one of the criteria was depressed mood or loss of interest or pleasure in daily activities.

### Classification and cleaning of screenable questions

Variable codes and descriptions were extracted from the publicly available codebook for the 2017 survey [16]. 1,440 variables having dichotomous response codes were manually identified and corresponding code values were denoted in a data contract file. Only codes corresponding to valid responses were selected for analysis; response codes indicating poor response quality or internally inconsistent responses were discarded. Additionally, the NSDUH includes several

**Table 1. Demographic summary of 2017 NSDUH responses.**

|  | Youth (ages 12–17) | Adults (ages 18+) | Total |
|---|---|---|---|
| **Respondents** | 13,722 | 42,554 | 56,276 |
| MDE in past year (≥5/9 DSM-5 criteria) | 1,814 (13.2%) | 3,949 (9.3%) | 5,763 (10.2%) |
| **Sex** | | | |
| Male | 7,050 (51.4%) | 19,987 (47.0%) | 27,037 (48.0%) |
| Female | 6,672 (48.6%) | 22,567 (53.0%) | 29,239 (52.0%) |
| **Ethnicity** | | | |
| White | 7,247 (52.8%) | 25,870 (60.8%) | 33,117 (58.8%) |
| Black/African American | 1,817 (13.2%) | 5,230 (12.3%) | 7,047 (12.5%) |
| Native American/Alaskan Native | 206 (1.5%) | 640 (1.5%) | 846 (1.5%) |
| Native HI/Pacific Islander | 65 (0.5%) | 195 (0.5%) | 260 (0.5%) |
| Asian | 561 (4.1%) | 2,070 (5.0%) | 2,631 (4.7%) |
| More than one race | 778 (5.7%) | 1,381 (3.2%) | 2,159 (3.8%) |
| Hispanic | 3048 (22.2%) | 7,168 (16.8%) | 10,216 (18.2%) |
| **DSM-5 MDE Diagnostic Criteria** | | | |
| Depressed mood for most of the day, almost every day | 1,721 (94.9%) | 3,807 (96.4%) | 5,528 (95.9%) |
| Loss of interest or pleasure in most things | 1,713 (94.4%) | 3,725 (94.3%) | 5,438 (94.4%) |
| Changes in appetite or weight | 1,468 (80.9%) | 3,587 (90.8%) | 5,055 (87.7%) |
| Problems with sleep | 1,722 (94.9%) | 3,822 (96.8%) | 5,544 (96.2%) |
| Others noticed that respondent was restless or lethargic | 918 (50.6%) | 2,128 (53.9%) | 3,046 (52.9%) |
| Felt tired and/or low energy nearly every day | 1,723 (95.0%) | 3,820 (96.7%) | 5,543 (96.2%) |
| Felt worthless nearly every day | 1,365 (75.2%) | 2,679 (67.8%) | 4,044 (70.2%) |
| Inability to concentrate or make decisions | 1,747 (96.3%) | 3,550 (89.9%) | 5,297 (91.9%) |
| Any thoughts or plans of suicide | 1,514 (83.5%) | 2,973 (75.3%) | 4,487 (77.9%) |

Demographic summary of study population by self-reported sex, ethnicity, and DSM-5 derived MDE diagnostic criteria. Sex and ethnicity percentages are calculated from the total number of respondents in the youth and adult cohorts; DSM-5 criteria percentages were calculated from the total number of respondents with MDE within the past year in each cohort.

age-dependent sets of questions that could not be compared across ages. These questions were excluded from analysis due to insufficient data in either the adult or the youth cohort.

## Survey-wide association analysis

The full dataset from the 2017 NSDUH was stratified into two groups on the basis of age: adults (18+ years) and youth (12–17 years). For each group, odds ratios [17] were calculated for each dichotomous variable against the presence of past-year MDE.

Fisher's exact test was used to assess the significance of association between each dichotomous variable and past-year MDE occurrence. Bonferroni correction with alpha = 0.01, n = 1,440 was set to control family-wise error rate (requiring p < 6.9e-6) as used in genome-wide association studies [14]. Bonferroni-corrected odds ratio (OR) 99.99% $(1 - \frac{\alpha}{2n})$ confidence intervals were calculated for the association between each variable and past-year MDE in each group (adult and youth) using the R fisher.test function [18].

Variables where the width of the Bonferroni-corrected confidence interval for the logarithm base 10 of the odds ratio (LOD) was greater than 3 were discarded as having insufficient durability for analysis. These criteria discard variables where the estimate for odds ratio varied over more than 3 orders of magnitude, including only results of high reproducibility for downstream analysis in this screening-based approach.

## Statistical tests

Variables with disjoint Bonferroni-corrected 99.99% confidence intervals for odds ratio estimates using the R fisher.test function were called as statistically significantly with alpha = 0.01, n = 1,440 [18]. To determine the degree of concordance between youth and adult associations, Lin's concordance correlation coefficient was calculated [19]. Additionally, a linear model was fit against odds ratio estimates for youth and adults using a least-squares error minimization approach.

## Additional analyses

After survey-wide association study was completed, hierarchical clustering was performed to enable better organization of results. Jaccard's distance index [20] was used to calculate a distance matrix for the variables called as significantly different. Complete linkage clustering was used to group variables by the computed distance matrix, and resultant dendrogram was partitioned using R language core tree utilities [18].

As an auxiliary descriptive analysis of the survey population, a smoothed kernel density plot of recalled age of first MDE for adult respondents with past-year MDE was computed using the default Gaussian kernel of the R ggplot2 library [21].

## Access to software

All code used to generate plots, and manual classification of variables analyzed, have been made available for public use, inspection, and improvement on GitHub under the MIT license [22].

# Results

After processing the 1,440 variables classified as dichotomous, 781 (54%) had a Bonferroni-corrected odds ratio confidence interval less than 3 orders of magnitude in width. The remainder of the variables were discarded, as insufficient data was present to estimate the odds ratio of these variables against past year MDE with a high degree of confidence in either the youth or the adult cohort. Reasons for insufficient data included questions that pertained only to youth or adult experience (e.g. school or work-related experiences) or a low number of respondents (e.g. heroin abuse in youth). Plotted against each other, the logarithm of odds ratios for variables with sufficiently narrow confidence intervals show high correlation (Fig 1). A linear model fit to the relationship between the adult and youth logarithm of odds ratio for all variables showed high correlation between youth and adult associations and high predictability for the model (slope = 0.95±0.02, intercept = -0.01±0.01, $R^2$ = 0.82, p < 0.001). Lin's concordance correlation coefficient between adult and youth associations was 0.91 (95% CI 0.89–0.92), showing moderate to excellent concordance between youth and adult odds ratios [23].

While most variables showed a high degree of concordance, 14 variables were found to have statistically significantly different association with youth and adult past-year MDE. Broadly, these differentially associated variables (DAVs) corresponded to: biological sex, Medicaid/CHIP status, tobacco use, drug and alcohol use, and characteristics of depressive episodes. While 3 variables were found to have a higher degree of association with adult MDE than youth MDE, the remaining 11 variables showed greater association with youth MDE (Fig 2). DAVs could be further segmented into variables where associations in the youth and adult cohorts were in the same direction but differed in strength (quantitative difference), and those where only a single cohort displayed significant association (qualitative difference). Qualitative differences were observed in variables tracking past-month alcohol use, Medicaid/CHIP status, marijuana use alone, and whether a single depressive incident could be picked out as the worst ever experienced.

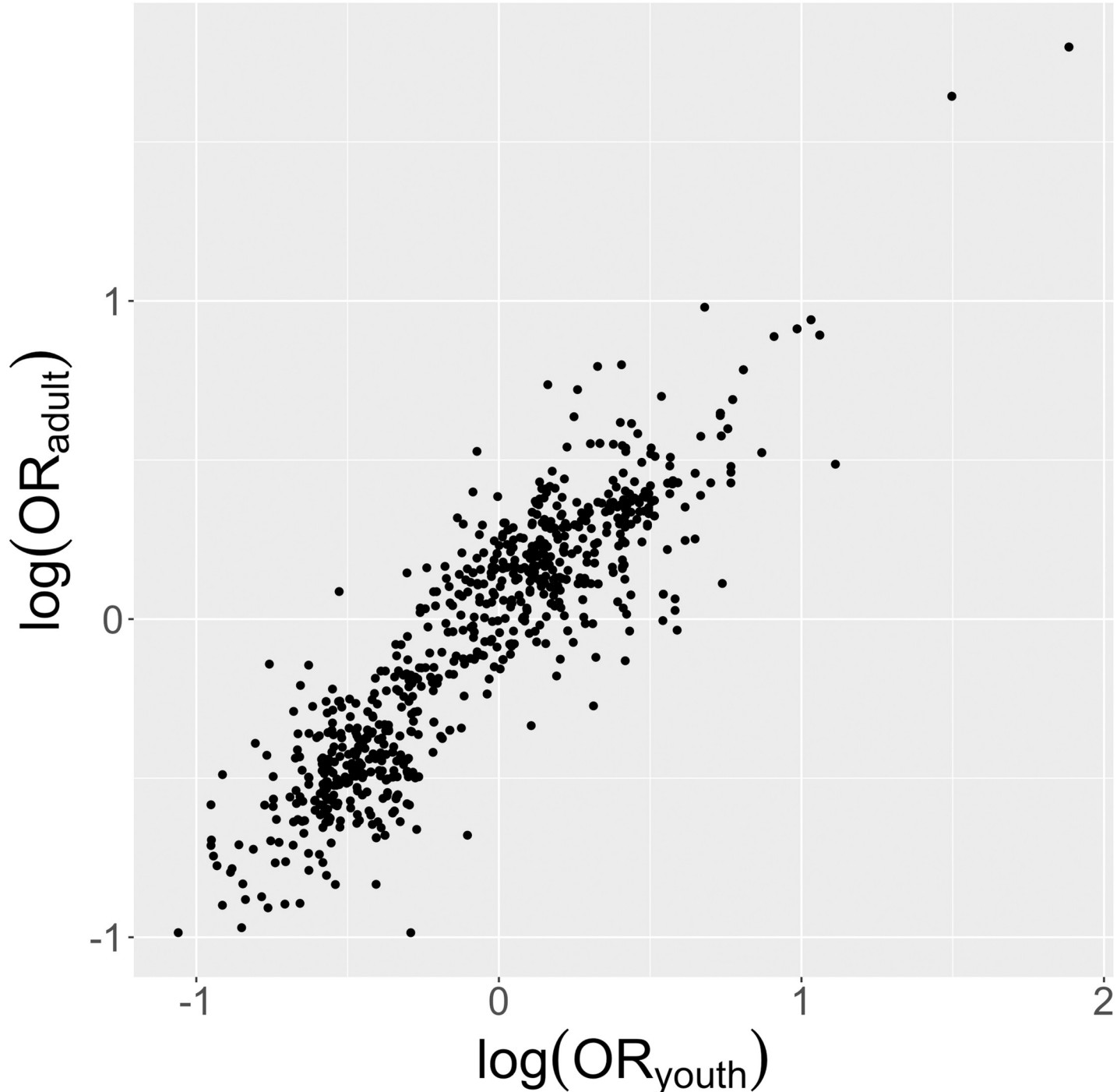

**Fig 1. Overview of calculated odds ratios.** Logarithm of the odds ratio for each variable against past-year MDE in the youth (aged 12–17 years) cohort plotted against logarithm of the odds ratio in the adult (aged 18+ years) cohort shows most variables similarly correlated. Variables where the size of the confidence interval was greater than 3 were discarded.

Reported difficulty concentrating, remembering or making decisions due to a physical, mental, or emotional condition had the greatest association among DAVs with both youth and adult MDE (OR = 4.79, p < 1e-10 and OR = 11.49, p < 1e-10 respectively) but was more

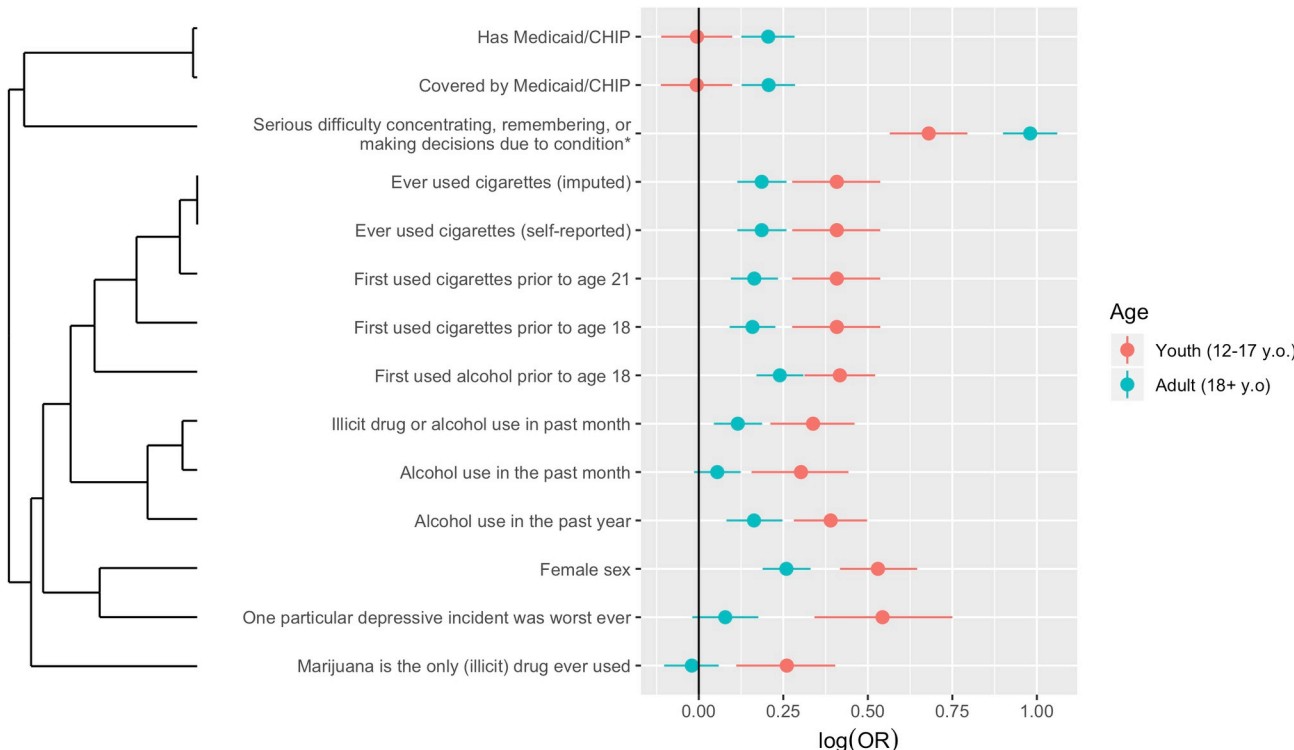

**Fig 2. Differentially associated variables in youth and adult depression.** Logarithm of the odds ratio for differentially associated variables with 99.99% confidence intervals. Hierarchical clustering generated by Jaccard distance index and complete-linkage clustering groups variables into related sets. *Condition was defined as either a physical, mental, or emotional condition.

associated with adults. Recalling a single depressive incident as the worst ever was more associated with youth than adults (OR = 3.50, p < 1e-10 and OR = 1.19, p = 1.9e-4). Several alcohol and tobacco use associated DAVs were identified, all with stronger MDE association in youth as compared to adults. Odds ratios estimates for cigarette-use related variables were closely clustered, ranging from 1.44 to 1.53 in adults as compared to 2.56 in youth (all p < 1e-10). Female sex was also more correlated with MDE in youth as compared to adults (OR = 3.39, p < 1e-10 and OR = 1.82, p < 1e-10 respectively). Alcohol use DAVs showed more spread in adults, with odds ratio estimates varying from 1.33 (p = 2.3e-4) to 1.77 (p < 1e-10), as compared to 2.01 to 2.61 in youth (all p < 1e-10). Marijuana use alone was associated with MDE in youth, but not in adults (OR = 1.8, p < 1e-10 and OR = 0.95, p = 0.24). Conversely, Medicaid / CHIP coverage was associated with MDE in adults but not in youth (OR = 1.61, p < 1e-10 and OR = 0.99, p = 0.80).

A Gaussian kernel density plot of recalled age of first MDE amongst adult respondents with past-year MDE shows a bimodal distribution for respondents greater than 26 years of age (Fig 3). In each age group, a peak is observed between 15 and 18 years old; a large portion of respondents with recent MDE recall their first MDE occurring during late youth.

## Discussion

While a high concordance rate was calculated between per-variable association with youth and adult past-year MDE, only a little over half of the dichotomous variables screened in the survey met inclusion criteria for analysis after Bonferroni multiple comparisons correction. Because of its novel approach to association screening in large surveys, this study remains conservative

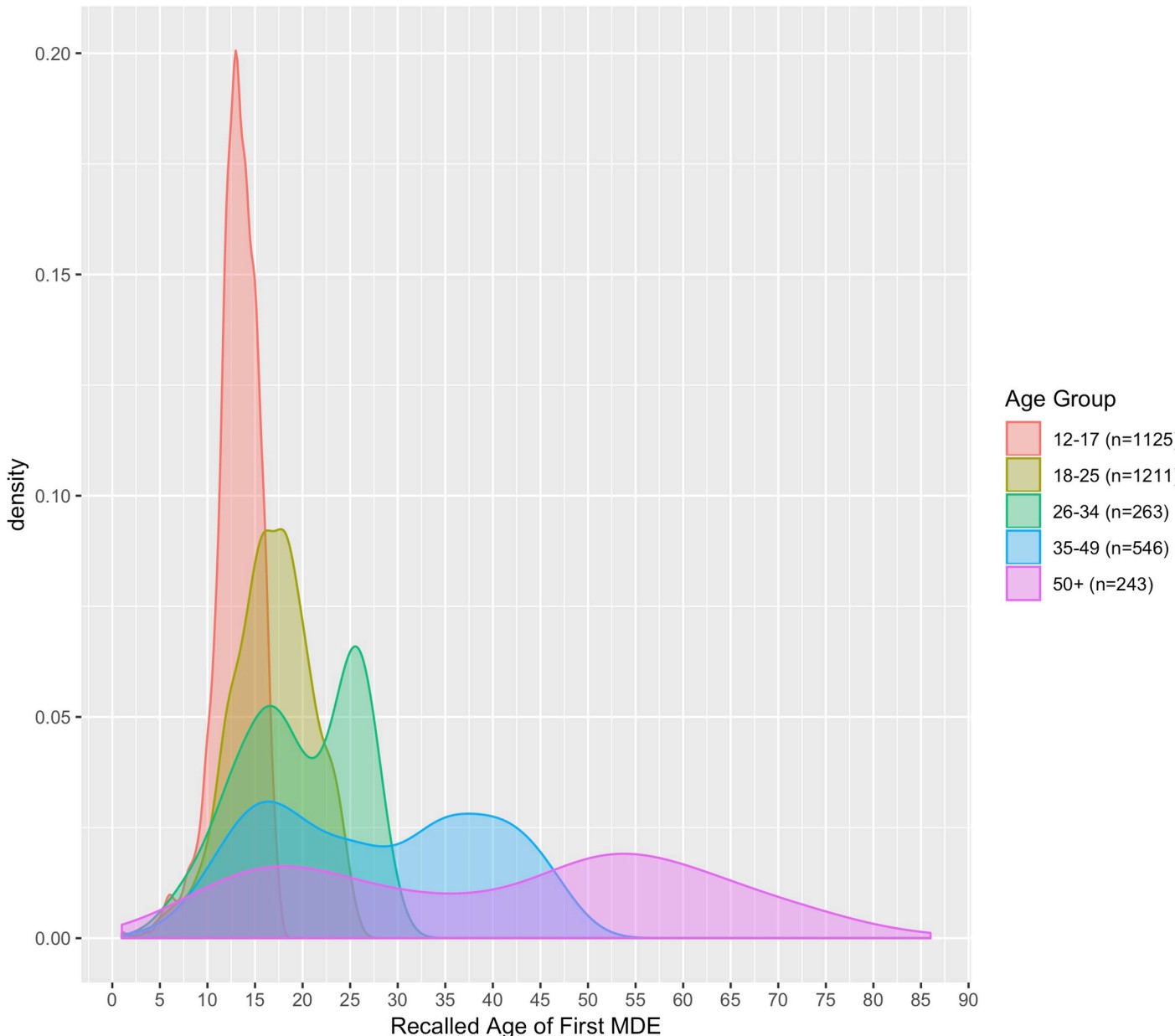

**Fig 3. Recalled age of first depressive episode by birth cohort.** Gaussian kernel density plot of age of first MDE in respondents with a past-year MDE segmented by age of respondents. A bimodal distribution is observed in 26–34, 35–49 and 50+ categories, with first MDE either in late youth (approximately 17 years old) or closer to respondent age.

with statistical bounds on generated estimates, while also weighing the dangers of exclusion bias [24]. While the variables that had sufficient measurements of responses to yield tight odds ratio estimates showed a general trend towards concordance in association between youth and adult past-year MDE, examination of these variables alone restricts this study's analysis to phenotypic parameters that are broadly applicable and for which both states measured are present in approximately equal frequency. At least one very significant known correlate of depression, suicide, does not meet these criteria, and would be poorly examined by this study design [25]. To generate more power for future analyses, data could be integrated across multiple years of

the NSDUH or across multiple disparate data sources. Merging data must be done carefully in the screening design to ensure that differences in survey semantics across years do not generate artefacts.

Despite the aforementioned limitations of this study's methodology, the analysis shows a high degree of similarity between youth and adult depression, at least among variables measured in the NSDUH. The calculated concordance coefficient of 0.91 (95% CI 0.89–0.92) and moderate to good predictability of a fitted linear model ($R^2 = 0.82$, $p < 0.001$) support this interpretation. These findings are consistent with previous descriptions of major depression as a disease where symptoms are mostly independent of patient age, but not entirely so [26–28]. Taken together, these data suggest that within the spectrum of data captured by dichotomous variables in the NSDUH, the phenotypic correlates of youth and adult depression are mostly similar. However, even with the strict multiple comparisons correction and significance criteria imposed by the survey-wide association procedure, some variables demonstrated clear differences by age and deserve further examination.

The differentially associated variables identified in this survey-wide association study could be grouped into a few key categories: female sex, Medicaid/CHIP status, alcohol use, cigarette use, marijuana use, cognitive symptoms, and perceived MDE severity. Of these, female sex, alcohol use, and cigarette use are well-described as correlates of depression in youth [29–35]. The association of marijuana use with MDEs in youths has some evidence, but is not as well-documented as the aforementioned correlates. Finally, this study identified novel correlates differentially associated with MDE in youth as compared with adults: Medicaid/CHIP status, cognitive symptoms (difficulty concentrating, remembering and making decisions), as well as the perception that one particular MDE was the worst ever experienced.

The effects of cannabis use have been an area of active interest as policymakers navigate changing regulations and social perceptions around use. Previous work in both humans and animal models found that cannabis exposure may have an effect on development in the brain–increased exposure is associated with reduced prefrontal and parietal gyrification in adolescents [36, 37]. Longitudinal MRI studies show that myelinogenesis and dendritic pruning drive changes in the adolescent brain from the onset of puberty to 24 years of age [38]. In addition to pharmacological effects, cannabis use may also function as an externalized indicator of social state. Social perception of cannabis has become increasingly positive in recent years; laws have passed in several U.S. states legalizing marijuana use for medical and recreational purposes [39]. Recent studies have also reported declining perception of risk associated with cannabis use [40]. As social perception and norms shift, a better understanding of the relative contribution of social and pharmacological factors will be needed to improve youth depression screening.

Healthcare coverage by Medicaid and/or the Children's Health Insurance Program (CHIP) showed qualitative changes in this study's screen; coverage is associated with adult MDE but not with youth MDE. Medicaid/CHIP jointly provide health services for low-income Americans and those with disabilities. Medicaid and CHIP's mental health services represent a significant portion of total spending in these programs, with Medicaid spending for enrollees with a behavior health diagnosis totaling approximately 85 billion USD in 2011 [41]. Medicaid and CHIP are programs intended to serve low-income patients known to be at increased risk for major depressive disorder in adulthood, as replicated by this study's data [42]. In the rich literature examining the links between familial socioeconomic and environmental factors with depression, complexity is the rule, and a clear association has not yet been described [43, 44]. While Medicaid/CHIP enrollment for children is not the ideal barometer for measuring socioeconomic status, the results suggest the absence of a robust relationship between childhood depression and familial socioeconomic status, and add to the dialogue in this debated area.

The symptom cluster of major depressive disorder is often characterized as consisting of affective, cognitive, and vegetative components, where affective symptoms are noted as potentially presenting atypically in adolescence [7, 45]. This study identified that a reported serious difficulty concentrating, remembering, or making decisions due to a physical, mental, or emotional condition was less associated with youth MDE than adult MDE. The symptoms measured by the variable are cognitive in nature, and may suggest underlying differences in presentation between cognitive symptoms in major depressive episodes in youth as compared with adults. However, when asked specifically about ability to think, concentrate, or make decisions during an MDE, youth reported impairment in slightly *greater* proportion (Table 1). Alternatively, the wide gap in association between youth and adults in the cognitive variable identified in this study's screen may be due to decreased attribution of cognitive changes to physical, mental or emotional conditions. Decreased attribution of changes in this manner may impact the psychometric properties of screening batteries for youth depression.

Finally, the finding that ability to pick out a single "worst" MDE was more associated with youth than adults demonstrates the need for careful assessment of variables for potential biases. This finding may suggest that youth depression follows a more turbulent course than adult depression, having more variation in symptoms apparent to patients. Major depressive disorder follows a course which often includes variation of symptoms over time, partial remission, and relapse [46]. In such a disease course, more exaggerated symptoms could occur during an age of peak onset such as adolescence. However, determining a single MDE as the worst may be affected by recall bias, and confounded by the number of MDE events experienced over the course of a respondent's lifetime. As older patients with longer disease course are more likely to have experienced more MDEs over a longer period of time, these factors should be corrected for before variable effect can be assessed. Unfortunately, as the public NSDUH dataset does not contain the exact age of each respondent, further studies with different datasets will be required to better assess this potential association while correctly correcting for confounders.

In addition to the primary survey-wide association study findings, a Gaussian kernel density plot for the recalled age of MDE onset was generated from NSDUH data to further characterize differences in depression presentation over the lifespan. The plot generated shows a strikingly consistent bimodal distribution, with peaks between 15 and 18 years old indicating that in the NSDUH dataset, a large proportion of respondents in each age cohort with a past-year MDE recall their first event as occurring during youth. Furthermore, co-location of peaks across groups suggests that, while subject to recall bias, these results are indicative of a true signal–that "adolescence" is a key time point in the progression of major depressive disorder [5]. Notably, the bimodal distribution identified in the NSDUH dataset is different from the fairly linear increase in depression prevalence reported in the epidemiological study cited by the DSM-5 [7, 47]. While interpretation of this study's results should be tempered by the potential for bias, they nevertheless support continuing exploration of phenotype in these disorders with an eye towards elucidating pathophysiology.

While the approach used by this study provides a novel strategy to identify associations, it also has significant limitations. Due to a lack of machine-readable data types for many surveys, the initial data curation requires labor-intensive manual classification. Complex organization of survey data requires correction for internal correlations as well as careful post-hoc analysis and grouping of variables identified by association techniques. The cross-sectional design of this study prevents identification of cause and effect relationships. Additionally, because this study does not semantically categorize variables prior to association screening, some variables identified may not be fully comparable between cohorts (e.g. cigarette use prior to age 18). As such, associations identified in this study's screen must be examined critically for possible

confounding factors and confirmed by independent investigations using hypothesis-based approaches. Finally, the need for strict false-discovery corrections reduces the power of an association-based study design, and some true associations may not have met the study's calling thresholds; absence of evidence should not be treated as evidence of absence. Despite these limitations in the interpretation of particular results, this study's survey-wide association screening approach allowed for examination over a much wider breadth of variables than typically interrogated. As data availability continues to improve and coding standards continue to develop, approaches like these will continue to become more viable and more valuable.

This study has utilized a survey-wide association approach to identify youth-specific correlates of major depressive episodes from data gathered in the 2017 NSDUH. By first codifying types and valid values for data gathered in the survey, this study's screen was able to confirm several previously described risk factors for youth depression and also to identify new ones. The new associations identified in marijuana use, cognitive impairment, and time-varying severity of disease may help to better define the pathophysiology of major depression in youth as compared with its adult incarnation. With a better understanding of disease classification and natural history, healthcare providers might be better equipped to deliver accurate prognoses and design effective treatments for patients both young and old with this common and debilitating condition.

## Acknowledgments

The author would like to thank Dr. Andrés Martin and Dr. James Leckman for their thoughtful discussions, support, and mentorship.

## Author Contributions

**Conceptualization:** Rahul Madhav Dhodapkar.

**Data curation:** Rahul Madhav Dhodapkar.

**Formal analysis:** Rahul Madhav Dhodapkar.

**Funding acquisition:** Rahul Madhav Dhodapkar.

**Investigation:** Rahul Madhav Dhodapkar.

**Methodology:** Rahul Madhav Dhodapkar.

**Project administration:** Rahul Madhav Dhodapkar.

**Resources:** Rahul Madhav Dhodapkar.

**Software:** Rahul Madhav Dhodapkar.

**Supervision:** Rahul Madhav Dhodapkar.

**Validation:** Rahul Madhav Dhodapkar.

**Visualization:** Rahul Madhav Dhodapkar.

**Writing – original draft:** Rahul Madhav Dhodapkar.

**Writing – review & editing:** Rahul Madhav Dhodapkar.

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
