## [Decision Letter · Decision Letter 0]

26 Mar 2020

PONE-D-20-05227

A survey-wide association study to identify youth-specific correlates of major depressive episodes

PLOS ONE

Dear Mr. Dhodapkar,

Thank you for submitting your manuscript to PLOS ONE. After careful consideration, we feel that it has merit but does not fully meet PLOS ONE’s publication criteria as it currently stands. Therefore, we invite you to submit a revised version of the manuscript that addresses the points raised during the review process.

We would appreciate receiving your revised manuscript by 16th April. To enhance the reproducibility of your results, we recommend that if applicable you deposit your laboratory protocols in protocols.io, where a protocol can be assigned its own identifier (DOI) such that it can be cited independently in the future. For instructions see: http://journals.plos.org/plosone/s/submission-guidelines#loc-laboratory-protocols

We look forward to receiving your revised manuscript.

Kind regards,

Marianna Mazza

Academic Editor

PLOS ONE

Journal Requirements:

2. Please correct the mention in the introduction that "depressive disorders affect more than 264 people worldwide" (line 43).

Reviewers' comments:

Reviewer's Responses to Questions

**Comments to the Author**

1. Is the manuscript technically sound, and do the data support the conclusions?

Reviewer #1: Partly

Reviewer #2: Partly

2. Has the statistical analysis been performed appropriately and rigorously? 

Reviewer #1: Yes

Reviewer #2: Yes

3. Have the authors made all data underlying the findings in their manuscript fully available?

Reviewer #1: Yes

Reviewer #2: Yes

4. Is the manuscript presented in an intelligible fashion and written in standard English?

Reviewer #1: Yes

Reviewer #2: Yes

5. Review Comments to the Author

Reviewer #1: This paper examines the correlates of major depressive episodes among youth and adults. Strengths include the rich data source and the helpful figures. Weaknesses include the incomplete description and justification of the methods and the exclusion of non-dichotomous correlates. Here are my specific comments and questions:

Introduction: The paper is clearly written. The discussion of prior literature is adequate. The benefits of the association design over other designs are unclear. Why couldn't "traditional methods" also screen a large number of variables? And what, specifically, does the author mean by "traditional methods"?

Methods: Why limit the predictors to only dichotomous variables? Were there no non-dichotomous variables that were worthy of examination? Is this a major disadvantage of the association method?

It is evident that the NSDUH surveyed minors as well as adults, but what specific ages were surveyed? How young were the youth? (This is reported below, but please report it in the sample description as well.)

The prevalences of the major depressive episode criteria seem incredibly high. Most respondents endorsed most of the items. How does this translate into relatively modest prevalence rates of MDEs? How were the items combined into the MDE variable?

The author uses the fairly conservative Bonferroni correction. Would conclusions about the differences between MDEs in youth and adults change if a less conservative but equally valid means of adjusting for multiple inferences were used?

Can the author say more about why only 781 variables were retained? In plain language, what was wrong with the others?

Reviewer #2: The author describes some interesting findings including some differential observations between variables associated with adult vs. adolescent MDE and a bimodal distribution of age of onset for MDD for those over 26 years of age. The paper is well written and clear. Overall it has the potential to be a valuable contribution I recommend that the following issues are attended to before publication:

Line 43: Please correct the number of people suffering w/ depressive disorders (currently written as 264 but I think it’s probably 264 million). Please also write out World Health Organization.

For many topics the NSDUH asks different questions of participants depending on age. E.g. many questions are asked of 12-17 year olds and are thus coded as missing or not applicable for adults. I think this is addressed briefly in the discussion (line 226) when you clarify that parameters must be present in approximately equal frequency in both samples to be considered but please clarify this in the methods section.

Also, I am not sure that some of the variables listed in figure 2 can be meaningfully compared between adults and adolescents. E.g. “first used cigarettes prior to age 18” and “first used cigarettes prior to age 21” would necessarily be coded as “yes” by default for anyone under the age of 17 who has ever smoked.

I also see that some variables which are not dichotomous such as race may have been dichotomized for the purpose of analysis. Is that accurate and if so was this done for any other variables?

The discussion of limitations is generally thorough but overall could benefit from a closer examination of the above issues.

6. PLOS authors have the option to publish the peer review history of their article (what does this mean?). If published, this will include your full peer review and any attached files.

Reviewer #1: No

Reviewer #2: No

---

## [Author Response · Author response to Decision Letter 0]

11 Apr 2020

# Response to Reviewers

## General Summary of Changes

We thank the reviewers and editorial staff for their time and the opportunity to submit a revised edition of our manuscript. 

Your feedback has helped to improve this manuscript and qualify our chosen approach. We have carefully considered the comments and responded to each point individually. In general, we have expanded some of the discussion around limitations of our association study approach and also clarified some ambiguous terms regarding methodology. We have updated specific information regarding the sample description and have uploaded additional code with expanded analysis to a publicly available repository.

We hope the specific responses below help address the points raised and thank the reviewers in advance for their time.

## Responses to Specific Comments

### Reviewer #1:

** Why couldn't "traditional methods" also screen a large number of variables? And what, specifically, does the author mean by "traditional methods"?

Thank you for this comment and also for the comments generally regarding the choice of the association study design. We agree that the term "traditional methods" was unnecessarily vague and does not capture our intent regarding our choice of association study design. By "traditional methods", we intended to denote hypothesis-driven case control studies. However, association studies are also a well-worn study design and could also be considered to be "traditional". While there is no technical barrier to using the hypothesis-driven approach to screen a large number of variables, doing so is often resource intensive and can be logistically difficult. We believe our approach may offer a scalable, cost-conscious adjunct to more exhaustive case-control methodology.

We simply wanted to highlight our goal of providing hypothesis-generating results, focusing on breadth, rather than depth. To further clarify this, we have added the following sentence to the text body and removed the reference to "traditional methods".

"While hypothesis-driven case-control studies carefully control for confounding and limit the number of exposures examined, our association study design hopes to provide a broader, albeit less granular, picture of disease-associated phenotypes to inform hypothesis testing. Our method serves as a low-cost, logistically expedient adjunct to hypothesis-driven studies; our analytical code and variable type classification have been made freely available and are ready for adaptation by other researchers."

** Methods: Why limit the predictors to only dichotomous variables? Were there no non-dichotomous variables that were worthy of examination? Is this a major disadvantage of the association method?

In our current study design, we limited our analysis to dichotomous variables to streamline our analytical process. By considering only this single class of variable, we were able to use odds ratio estimates and multiple-comparisons-corrected confidence intervals for all variables, consistent with the type of association study framework that is used in genome-wide association studies. By mirroring the genome-wide association study approach and borrowing some best practices regarding multiple-comparisons corrections, we hoped to limit the number of false positive findings that could arise from scanning all variables within a large survey.

While there is no ostensible disadvantage to using alternative statistical approaches within the association study framework to incorporate non-dichotomous variable types, there is a paucity of precedent to support such an expansion. 

Our methodology might be expanded to include other measures such as continuous-time analysis or associations between Likert-type variables, both of which are relevant to common data types present in the large-scale surveys we hope to mine. Many of these non-dichotomous variables will certainly contain correlates worthy of examination.

To better address these concerns in the main body of our paper, we added the following:

"This study screened only dichotomous variables to streamline statistical analysis within the association study framework, but we hope that in the future our approach can be expanded to include additional data types such as continuous time or Likert-type variables."

** It is evident that the NSDUH surveyed minors as well as adults, but what specific ages were surveyed? How young were the youth? (This is reported below, but please report it in the sample description as well.)

Thank you for your comment; we have included the ages surveyed (12-17 for youth and 18+ for adult) in the sample description table as suggested. Additionally, the sample description was amended with the sentence: "The survey included both youths (12-17) and adults (18+) in its direct data collection."

** The prevalences of the major depressive episode criteria seem incredibly high. Most respondents endorsed most of the items. How does this translate into relatively modest prevalence rates of MDEs?

The rates of depressive episode criteria reported in the sample description table are the rates of these criteria within the subset of respondents that was classified as having a major depressive episode within the past year. The percentages reported reflect this, as noted in the legend for Table 1, hence they are very large as you mentioned (for the top two criteria, 90% plus). These items were provided to give a clinical sense of the respondents who were classified as having a major depressive episode in both the youth and adult cohorts.

** How were the items combined into the MDE variable?

The items were combined into the MDE variable if a respondent reported having at least 5 out of the 9 criteria listed in the sample table (Table 1) where at least one of the criteria was depressed mood or loss of interest or pleasure in daily activities. This is equivalent to the Diagnostic and Statistical Manual 5th Edition (DSM-5) criteria for a major depressive episode. To better clarify this, we have included the following sentence in the description of the study population:

"Occurrence of past-year MDE was identified on the NSDUH if a respondent reported having at least 5 out of the 9 symptomatic criteria (Table 1) where at least one of the criteria was depressed mood or loss of interest or pleasure in daily activities."

** The author uses the fairly conservative Bonferroni correction. Would conclusions about the differences between MDEs in youth and adults change if a less conservative but equally valid means of adjusting for multiple inferences were used?

Yes, our conclusions could change if less conservative means of adjusting for multiple inferences were used. We prototyped using a less conservative approach (Benjamini and Hochberg correction) with false discovery rate q = 0.05 and generated 73 distinct variables with non-overlapping confidence intervals between youth and adult (as compared with 14 by Bonferroni correction). We will include this prototype code in our public repository containing all of our analysis to increase availability to other researchers. However, due to limitations of the association study design regarding false positive findings, we believe that the conservative approach is most appropriate for the main body of the text.

** Can the author say more about why only 781 variables were retained? In plain language, what was wrong with the others?

We thank the reviewer for raising this point of uncertainty. To address these concerns, we added the following to the manuscript to explain why only 781 variables were retained for downstream analysis, and to expand on the description regarding limiting the width of the confidence intervals:

"After processing the 1,440 variables classified as dichotomous, 781 (54%) had a Bonferroni-corrected odds ratio confidence interval less than 3 orders of magnitude in width. The remainder of the variables were discarded, as insufficient data was present to estimate the odds ratio of these variables against past year MDE with a high degree of confidence in either the youth or the adult cohort. Reasons for insufficient data included questions that pertained only to youth or adult experience (e.g. school or work-related experiences) or a low number of respondents (e.g. heroin abuse in youth)."

### Reviewer #2:

** Line 43: Please correct the number of people suffering w/ depressive disorders (currently written as 264 but I think it’s probably 264 million). Please also write out World Health Organization.

Thank you for your comments; both of these changes have been made in the manuscript.

** For many topics the NSDUH asks different questions of participants depending on age. E.g. many questions are asked of 12-17 year olds and are thus coded as missing or not applicable for adults. I think this is addressed briefly in the discussion (line 226) when you clarify that parameters must be present in approximately equal frequency in both samples to be considered but please clarify this in the methods section.

We agree that this should be clarified in the methods section. We added the following sentence to the methods section to cover this point:

"Additionally, the NSDUH includes several age-dependent sets of questions that could not be compared across ages. These questions were excluded from analysis due to insufficient data in either the adult or the youth cohort."

** Also, I am not sure that some of the variables listed in figure 2 can be meaningfully compared between adults and adolescents. E.g. “first used cigarettes prior to age 18” and “first used cigarettes prior to age 21” would necessarily be coded as “yes” by default for anyone under the age of 17 who has ever smoked.

We thank the reviewer for raising this point, as it highlights a limitation of our study design relating to confounding. While it is true that all respondents in the youth cohort who have ever smoked will have "first used cigarettes prior to age 18", the association between this exposure and major depressive episode (MDE) may be due to the smoking itself and not due to the "before age 18" component.

Because we do not know what sorts of confounders may be present in the data set, we provide our analysis and figures without removing these variables as a way of allowing readers to decide what types of associations may be present. They may then design experiments that independently validate these observations. As we show in our discussion, many of the correlates identified using our association protocol have been independently identified in prior literature. The reproduction of these existing findings strengthens our hope that our novel descriptions may be used as a starting point for further research.

We have included a sentence in the discussion section to further address this point:

"Additionally, because we do not semantically categorize variables prior to association screening, some variables identified may not be fully comparable between cohorts (e.g. cigarette use prior to age 18). As such, associations identified in our screen must be examined critically for possible confounding factors and confirmed by independent investigations using hypothesis-based approaches."

** I also see that some variables which are not dichotomous such as race may have been dichotomized for the purpose of analysis. Is that accurate and if so was this done for any other variables?

Only variables which were coded as dichotomous in the NSDUH Codebook were analyzed as dichotomous. Some non-dichotomous demographic variables such as race were independently compiled for the purposes of Table 1, so that readers could have a clear view of the breadth of coverage of the NSDUH sample population, and to provide a first-order demographic summary of the youth and adult cohorts as they compared to each other. No non-dichotomous variables were dichotomized by the authors during the analysis.

** The discussion of limitations is generally thorough but overall could benefit from a closer examination of the above issues.

We thank the reviewer for this feedback. We hope that the above additions help to clarify these concerns.

---

## [Editor Report · Decision Letter 1]

14 Apr 2020

A survey-wide association study to identify youth-specific correlates of major depressive episodes

PONE-D-20-05227R1

Dear Dr. Dhodapkar,

We are pleased to inform you that your manuscript has been judged scientifically suitable for publication and will be formally accepted for publication once it complies with all outstanding technical requirements.

With kind regards,

Marianna Mazza

Academic Editor

PLOS ONE
---

## [Editor Report · Acceptance letter]

30 Apr 2020

PONE-D-20-05227R1 

A survey-wide association study to identify youth-specific correlates of major depressive episodes 

Dear Dr. Dhodapkar:

I am pleased to inform you that your manuscript has been deemed suitable for publication in PLOS ONE. Congratulations! Your manuscript is now with our production department. 

With kind regards,

on behalf of

Dr. Marianna Mazza 

Academic Editor

PLOS ONE